# Analysis of Genetic Variants in the Glucocorticoid Receptor Gene *NR3C1* and Stenosis of the Carotid Artery in a Polish Population with Coronary Artery Disease

**DOI:** 10.3390/biomedicines10081912

**Published:** 2022-08-07

**Authors:** Jarosław Gorący, Anna Gorący, Aldona Wójcik-Grzeszczuk, Iwona Gorący, Jakub Rosik

**Affiliations:** 1Independent Laboratory of Invasive Cardiology, Pomeranian Medical University, 70-111 Szczecin, Poland; 2Department of Clinical and Molecular Biochemistry, Pomeranian Medical University, 70-111 Szczecin, Poland; 3Department of Chemistry, The University of Chicago, Chicago, IL 60637, USA

**Keywords:** carotid artery stenosis, atherosclerosis, *NR3C1* gene

## Abstract

**Background:** Cardiovascular diseases (CVDs) are the leading cause of death worldwide. Early diagnosis and elimination of risk factors are crucial for better managing CVDs. Atherosclerosis, whose development might be associated with glucocorticoids (GCs), is a critical factor in the development of carotid artery (CA) stenosis and most other CVDs. **Aim:** To investigate the association of Tth111I, N363S, and ER22/23EK-*NR3C1* polymorphisms and the incidence of CA stenosis. **Methods:** The study group consisted of 117 patients diagnosed with coronary artery disease (CAD) and CA stenosis and 88 patients with CAD and ruled out CA stenosis. Genomic DNA was extracted from blood, and genotyping was carried out using Tth111I, N363S, and ER22/23EK-*NR3C1* polymorphism sequencing. **Results:** No significant association between studied polymorphisms and the incidence or the severity of CA stenosis in the Polish population with CAD was found. **Conclusion:** This is the first study that proves that common *NR3C1* gene variants do not influence CA stenosis and probably are not associated with atherosclerosis. The search for genes that can act as prognostic markers in predicting CA stenosis is still ongoing.

## 1. Introduction

Cardiovascular diseases (CVDs) continuously remain the leading cause of death worldwide and are bound to increase as a burden with the aging population [1,2]. According to the World Health Organization (WHO), approximately 17.9 million people died from CVDs in 2019, representing 32% of global deaths [3]. The vast majority of these disorders would be better managed if early diagnosis and elimination of some common risk factors, including physical inactivity, unhealthy diet, and tobacco or alcohol consumption, were applied [4]. As a dominant cause of CVD, atherosclerosis is responsible for most cardiovascular morbidity and mortality. It is acknowledged as a chronic inflammatory disease resulting in the deposition and accumulation of endogenously modified structures, specifically oxidized lipoproteins beneath the endothelial layer of arterial walls [5]. The lipid lesions can be found in the aorta in the first decade of life, however, they may become the cause of symptoms during later years of the disease [6]. Genetic as well as environmental factors festinate the expansion of the atherosclerotic plaques so that they can be found in various arterial territories such as carotid, cerebral, coronary, or lower limb vessels [7]. Among people aged 30–79 years, approximately 28% have an abnormal carotid intima-media thickness, nearly 21% carotid plaque, and 1.5% carotid stenosis [8,9]. As atherosclerosis is a diffuse disease, patients with stenosis of the carotid artery (CA) are more expected to have atherosclerotic disease in other arteries [8,9]. The extensive group of people with carotid plaque or carotid stenosis indicates a potential for cerebrovascular events resulting in a massive load of diseases in the global healthcare system [8].

Glucocorticoids (GCs) are one of the numerous factors determining CVD pathogenesis. They are steroid hormones synthesized and released by the adrenal cortex in a circadian manner and as a response to stress. GCs are also used as therapeutic agents. Their side effects, such as abdominal obesity, diabetes, and arterial hypertension (HT), are simultaneously CVDs risk factors [10,11,12].

GCs act via the GC receptor (GR), a member of the nuclear receptor superfamily of ligand-dependent transcription factors. GR is encoded by the gene *NR3C1* located on chromosome 5q31–32 in humans [13]. *NR3C1* gene genetic variation may contribute to higher body mass index (BMI), elevated plasma lipid levels, insulin resistance, and HT, which are considered atherosclerosis risk factors [14,15]. However, some reports have shown contradictory results [14,16,17]. GR polymorphisms such as ER22/23EK and GR-9β have been associated with GC resistance syndrome, resulting in a better metabolic profile, including decreased total cholesterol levels [18,19]. N363S and *Bcl*I polymorphisms’ carriers have higher BMI and the risk of visceral obesity probably due to GC hypersensitivity [18]. Patients with A3669G polymorphism of the GR gene, which has been connected with GC resistance, have been reported to be at elevated risk of enlarged heart, systolic dysfunction, CAD, and heart failure (HF) [20,21].

In this study, we discussed the possible influence of the genetic polymorphisms Tth111I, N363S, and ER22/23EK of the GR gene *NR3C1* on the incidence of CA stenosis in a group of Polish patients subjected to carotid angiography.

## 2. Patients and Methods

### 2.1. Patients

The protocol of the study was approved by the Pomeranian Medical University Ethics Committee, with formal informed consent signed by all the participants.

We defined case group (CS) as patients with angiographically documented CAD admitted to the Cardiology Clinic due to CA stenosis. The program enrolled 117 consecutive patients aged 56–90 (76 men and 41 women). This population was further divided into a subgroup with 70–89% stenosis (CS70), including 61 patients (40 men and 21 women), and a second subgroup with critical (≥90%) stenosis (CS90), including 56 patients (36 men and 20 women). All patients enrolled in the program underwent percutaneous carotid angioplasty with stent implantation (CAS).

The control group (CG) consisted of 88 people aged 40–81 (35 men and 53 women) admitted to the hospital with non-specific chest pain and underwent coronary angiography to exclude CAD. The control group also had an ultrasound examination of the CAs, which ruled out the presence of CA stenosis.

HT was defined as either systolic blood pressure exceeding 140 mmHg, or diastolic blood pressure greater than 90 mmHg, or antihypertensive treatment. Subjects whose fasting plasma glucose was higher than 125 mg% (6.9 mmol/L) or those using antidiabetic medication were recognized as patients with diabetes mellitus (DM). Dyslipidemia was diagnosed when at least one of the following criteria was met:-Low-density lipoprotein cholesterol (LDL-c) 115 mg% or higher;-Triacylglycerols (TG) 150 mg% or higher;-High-density lipoprotein cholesterol (HDL-c) below 40 mg% (men) or below 45 mg% (women).

LDL-c, TG, and HDL-c were measured using enzymatic methods (Roche Diagnostics, Poland). BMI was calculated as weight/height^2^ (kg/m^2^). Patients were classified as current smokers if they reported a daily rate of more than five cigarettes. Patients who reported quitting smoking for five years or who did not smoke at all were classified as nonsmokers.

CA ultrasonography (USG) Doppler was performed according to standard procedures using GE VIVID E9 device with linear transducer 11L-D. The procedure was conducted according to standard procedures recommended by the American Society of Echocardiography (ASE) and Mannheim carotid intima-media thickness consensus [22]. They define atherosclerotic plaque as a focal thickening of the arterial wall > 50% in comparison to surrounding Intima-Media complex or Intima-Media complex thickness > 1.5 mm [23].

### 2.2. Genotyping

Genomic DNA was extracted from peripheral blood leukocytes with the QIAamp DNA Mini Kit (QIAGEN, Hilden, Germany).

For the identification of polymorphisms, the Tth111I [C(-3807)T], the ER22/ER23 (G198A/G200A), and the N363S (A1220G) *NR3C1* gene the method of sequencing with the following pairs of primers was used, respectively, F: 5′- TATTTGTTGGGTGCCTGCTATGTA -3′, R: 5′- GATGAACTCCAGTGTGCCAGAAAG -3′; F: 5′- GGCACAGTTTACTGTCAGGC-3′, R: 5′- CACTGATCTTACCTTGAATAGCCAT- 3′; and

F: 5′- GCTGCCTCTTACTAATCGGATCAG-3′, R: 5′- GCTGCTTGGAGTCTGATTGAG—3′. Amplification was performed on a Veriti Dx 96-Well Thermal Cycler (Thermo Fisher Scientific, Waltham, Massachusetts, USA). Analysis results were read with Sequencing Analysis v5.1 software (Applied Biosystems, Waltham, Massachusetts, USA).

### 2.3. Statistical Analysis

Quantitative variables were compared between groups using Mann–Whitney tests. χ^2^ or Fisher’s exact test was used for qualitative variables (genetic data, gender, presence of DM, HT, dyslipidemia, and stenosis). An exact test was applied to assess the conformity of the genotype distribution to Hardy–Weinberg law. *p* ≤ 0.05 was considered statistically significant.

## 3. Results

Baseline characteristics of CS, CS70, CS90, and CG are shown in Table 1. Frequency of male gender, DM, HT, and age were proven to be significantly higher among CS when compared to CG (*p* = 0.0004, *p* < 0.00002, *p* < 0.00002, and *p* < 0.00002, respectively). Statistically significant differences between CS70 and CS90 groups were not found.

The Tth111I, N363S, or ER22/23EK-*NR3C1* genotypes were found to be in Hardy–Weinberg equilibrium in both CS and CG (*p* > 0.05). The three loci were then examined for the level of LD in the entire group (*n* = 205). The estimated values of D’ were equal to 1 for each pair of loci.

The results of an association of the Tth111I, N363S, or ER22/23EK polymorphisms of the *NR3C1* gene with CA stenosis are summarized in Table 2. We did not observe any significant association between the occurrence of the disease and the variants tested. No significant association between Tth111I, N363S, or ER22/23EK-*NR3C1* polymorphisms and the severity of CA stenosis were found when CS70 vs. CS90 were compared (Table 3). Moreover, no significant associations were found among Tth111I, N363S, or ER22/23EK alleles or haplotypes and the occurrence or severity of CA stenosis (Table 2 and Table 3).

In addition, we conducted a separate analysis to compare subgroup hypertensive (CS&HT) and normotensive patients with CA stenosis (CS&nHT). The association between Tth111I, N363S, or ER22/23EK polymorphisms and HT did not reach statistical significance (Table 4).

Moreover, we tried to reduce the influence of sex hormones. Therefore, we conducted separate analyses to compare men with male patients without CA stenosis (Table 5) as well as women with and without CA stenosis (Table 6). No association reaching statistical significance was found.

Association among several metabolic profile parameters and the Tth111I, N363S, and ER22/23EK variants was assessed (Table 7). We did not observe any significant association between studied polymorphisms and BMI, dyslipidemia, DM, or HT.

## 4. Discussion

This study assessed the role of the Tth111I, ER22/ER23, and N363S-*NR3C1* gene polymorphisms in the development of CA stenosis in the Polish population with CAD. We have not shown any connection between CA stenosis and the genetic variants, alleles, or haplotypes of *NR3C1* gene. To the best of our knowledge, our study is the first to investigate the association between Tth111I, ER22/ER23, and N363S-*NR3C1* gene polymorphisms and the CA stenosis. The strength of our study is determined by the very well documented homogeneous cohort of individuals undergoing angiography balanced with the control group. The limitation of this research is the relatively small study group. Additionally, the control group was younger than the CS group, which may have had some impact on the obtained results. Certainly, another limitation is the lack of determinations of inflammatory factors (including GCs) in the blood serum. Moreover, both CS and CG have already been treated for CAD, which could modulate, or even to some extent, extinguish the inflammatory process in the vessels [24].

Atherosclerosis, stenosis, or contraction of the CA decreases brain perfusion, potentially causing ischemic stroke. Thus, atherosclerosis in the CA is a crucial factor in the pathobiology of brain symptoms and is considered an independent risk factor for cerebral events.

A complex interaction between the environmental factors, genetic components, and inflammation leads to the initiation and progression of atherosclerosis [5]. Particular attention should be paid to the inflammatory process. Despite researchers’ efforts, its exact mechanism is still not entirely clear. In the large multiethnic study, the links between none of 1348 *ALOX5* (arachidonate 5-lipoxygenase, 5-LOX) polymorphisms and subclinical atherosclerosis, as well as CAD events, were found [25]. Polymorphisms in this gene increase the inflammatory production of leukotrienes. It may contribute to the promotion of inappropriate or excessive vasculitis, leading to atherosclerosis and potential CVD [26,27,28]. Another study reported the supposed modifying effect of a diet rich in omega-3 and -6 fatty acids on the relationship between ALOX5 tandem variability and atherosclerosis [29]. However, in the multiethnic study by Tsai et al., such a relationship has not been confirmed [25].

In the study on the Netherlands population, Van Rossum et al. found that the ER22/23EK polymorphism of the GR gene was associated with lower C-reactive protein (CRP) levels and better survival rate in older men. This can be related to subtle lifelong GC resistance [30]. In contrast, a relatively large study of the South-West Germany population with weakly presented standard risk factors investigated the association between 16 pro-inflammatory genes and the incidence of stroke. It has been shown that the more pro-inflammatory genetic profile, the higher risk of stroke. Furthermore, some polymorphisms in the CRP gene, which cause its overexpression, are likely to increase the risk of CAD and myocardial infarction (MI) [31,32]. However, in our previous study of the Polish population, no connection was discovered between polymorphisms of pro-inflammatory cytokines such as IL1B C(-31)T/IL1RN (VNTR) or their haplotypes and CAD [33,34]. Although numerous studies suggest that inflammatory cytokines and their genetic polymorphisms might play an essential role in the pathogenesis of atherosclerosis and CA stenosis, the results remain controversial.

The harmful effect of GCs on peripheral tissues metabolism is well understood and has been the subject of intensive research for many years. The connection between GC sensitivity and increased risk of CVDs has been suggested in various studies [35,36,37]. It has been reported that GCs may trigger or modulate atherosclerosis, i.e., through regulating lipid metabolism [4]. GCs exert effects on the vasculature by GR regulation of various signaling pathways. The well-characterized impact of GCs on the vascular system include those mediated by GR modulation of NO biosynthesis [38]. Hence, variants of the *NR3C1* gene may affect the development and progression of CA atherosclerosis [4].

So far, several *NR3C1* gene polymorphisms have been described to affect the variable sensitivity of GCs and changes in the metabolic parameters [4]. The increased sensitivity to GCs leads to the chronic hyperactivity of the hypothalamic-pituitary-adrenal (HPA) axis, which harmfully affects the vascular system. It may contribute to atherosclerosis [39].

The most widely studied polymorphisms are ER22/23EK, BclI, GR-9β, N363S, and Tth111. In our study, no associations between traditional risk factors for cardiovascular diseases such as plasma lipid levels, DM, HT, or BMI and Tth111I, N363S, or ER22/23EK-*NR3C1* gene polymorphisms were found. Our results remain in accordance with other studies. Dobson et al. did not show any connection between 363S allele and BMI, plasma lipid levels or glucose tolerance status in the Caucasian population [17]. Koeijvoets et al., in a population with familial hypercholesterolemia that presented very high risk of CVD, observed no significant association with ER22/23EK polymorphism as well [40]. Moreover, another study conducted in the Dutch and British populations demonstrated a lack of association between *NR3C1* gene polymorphisms and risk factors such as HT, DM, or plasma insulin and cholesterol levels [17,41]. However, it should be noted that the traditional risk factors presented in these studies were not as strongly expressed as in the current study.

Contrariwise, it has been reported that Tth111I, N363S, or ER22/23EK-*NR3C1* gene polymorphisms are correlated with the development of CAD, atherosclerosis, and altered metabolic profile [21,42]. In the Turkish population, the N363S polymorphism of the GR gene was associated with obesity and higher weight and BMI in patients with DM type 2 (DM2) [43]. There is evidence of linkage between Tht111I and ER22/23EK polymorphisms with decreased GC sensitivity, lower LDL-c, and lower fasting insulin levels [44]. Moreover, van Rossum et al. proved that the ER22/23EK polymorphism (heterozygous for the 22/23EK) is associated with a better metabolic profile (lower LDL-c, insulin, and fasting glucose levels), leading to lowering the risk of DM2 and CVD. The effect in this case may be due to the reduced GC sensitivity [19]. In another large study, the BclI polymorphism of the GR gene relates to significantly greater total body fat, contributing to increased insulin resistance. The authors support the hypothesis that even slight genetic changes in the GR gene, which are known to alter the sensitivity of GCs, may have subtle metabolic implications [45]. The discrepancy between results of studies on the aforementioned polymorphisms’ influence on metabolic findings might be explained by the age differences or type 1 error that is a false-positive finding. However, it seems that these subtle changes may be mitigated when heavily influenced by environmental factors and dependent on the population.

In our previous study, an association between homozygous Tth111I and multivessel CAD was found for the first time to the best of our knowledge [46]. It is worth emphasizing that in the Polish population, traditional risk factors, such as unhealthy diet, smoking, or unfavorable lifestyle, are still strongly expressed and may obscure the influence of genetic factors. Therefore, the results still require further research in another population.

Despite the existence of strongly expressed environmental risk factors, the progression of atherosclerosis in the coronary vessels may differ from the CA. These mechanisms are still not fully understood. Alevizaki et al. conducted a study investigating the connection between the activity of the HPA axis and the severity of CVD in the Greek population [47]. Morning serum cortisol and intima-media thickness (IMT) levels were measured. It was proven that the GG homozygotes of the Bcl1 polymorphism of the GR gene had a much higher IMT in the CA. On the basis of these preliminary observations, it is suggested that both HPA axis hyperactivity and tissues’ hypersensitivity to GCs can independently contribute to the aggravation of atherosclerosis in the coronary and peripheral arteries [47]. However, other studies do not support this finding [48,49]. This might suggest that genetic variants of GR may be involved in the development and progression of atherosclerosis. However, one must consider the dissimilarities in different populations, the severity of environmental risk factors for atherosclerosis, and gene–gene and gene–environment interactions.

It has been shown that in different populations with varying intensity of traditional risk factors, the impact of genetics may have a variable extent [50,51]. Favé et al. have reported that local environment may have a direct impact on the disease risk phenotypes. Furthermore, the genetic variations, including less common variants, can also modulate individual responses to environmental challenges [52].

Although we did not find any association between *NR3C1* gene polymorphisms and CA stenosis, the search for genes that can act as prognostic markers in the prediction of poor outcomes and aid in selecting appropriate therapeutic intervention is still ongoing.

## 5. Conclusions

We did not find any association between *NR3C1* gene polymorphisms and CA stenosis either when other CA stenosis risk factors were included or excluded. However, the search for genes that can act as prognostic markers in the prediction of poor outcomes and aid in selecting appropriate therapeutic intervention is still ongoing.

## Figures and Tables

**Table 1 biomedicines-10-01912-t001:** Baseline characteristics of CG and CS.

Parameter	CG (*n* = 88)	CS (*n* = 117)	*p* CG vs. CS^a^	CS70 (*n* = 61)	CS90 (*n* = 56)	*p* CS70 vs. CS90 ^a^
Male gender	39.77%	64.96%	0.0004	65.57%	64.29%	1.00
Age years	60.14 ± 8.25	72.61 ± 7.96	<0.00002	72.98 ± 7.24	72.20 ± 8.66	0.656
BMI (kg/m^2^)	27.40 ± 3.85	27.74 ± 3.80	0.674	28.07 ± 3.90	27.39 ± 3.66	0.304
Dyslipidemia	73.86%	64.96%	0.223	62.30%	67.86%	0.565
DM	14.77%	49.57%	<0.00002	49.18%	50.00%	1.00
HT	54.55%	86.32%	<0.00002	85.25%	87.50%	0.792
Smoking	12.50%	12.82%	1	11.48%	14.29%	0.784

CG—control group; CS—patients with CA stenosis; CS70—patients with 70–89% stenosis; CS90—patients with 90% or more stenosis; DM—diabetes mellitus; HT—arterial hypertension. Data are given as mean ± SD or %. ^a^ *p*-values were calculated with Mann–Whitney test for quantitative variables and with Fisher’s exact test for qualitative variables.

**Table 2 biomedicines-10-01912-t002:** Tth111I, N363S, and ER22/23EK genotypes in CG and CS groups.

Polymorphism	CG (*n* = 88)		CS (*n* = 117)		CG vs. CS			
	*n*	%	*n*	%	*p* ^a^	Compared Genotypes or Alleles	OR (95% CI)	*p* ^b^
Tth111I genotype					0.955			
CC	40	45.45%	54	46.15%		CC + CT vs. TT	0.85 (0.258–2.875)	0.792
CT	41	46.59%	55	47.01%		CC vs. CT + TT	0.972 (0.537–1.758)	1
TT	7	7.95%	8	6.84%		CC vs. TT	0.848 (0.246–2.994)	0.786
Tth111I allele								
C	121	68.75%	163	69.66%		C vs. T	0.958 (0.614–1.499)	0.914
T	55	31.25%	71	30.34%				
N363S genotype								
AA	82	93.18%	110	94.02%		AA vs. AG + GG	0.87 (0.24–3.261)	1
AG	6	6.82%	6	5.13%				
GG			1	0.85%				
N363S allele								
A	170	96.59%	226	96.58%		A vs. G	1.003 (0.299–3.577)	1
G	6	3.41%	8	3.42%				
ER22/ER23EK genotype								
AG	2	2.27%	1	0.85%		AG vs. GG	2.685 (0.138–160.345)	0.578
GG	86	97.73%	116	99.15%				
ER22/ER23EK allele								
A	2	1.14%	1	0.43%		A vs. G	2.672 (0.138–138.519)	0.579
G	174	98.86%	233	99.57%				
Tth111I/N363S/ ER22/23EK haplotype								
CAG	114	64.77%	156	66.67%		CAG vs. other	0.92 (0.597–1.419)	0.752
TAG	54	30.68%	69	29.49%		TAG vs. other	1.058 (0.675–1.656)	0.828
CGG	6	3.41%	7	2.99%		CGG vs. other	1.144 (0.312–4.056)	1
TAA	1	0.57%	1	0.43%		TAA vs. other	1.33 (0.017–104.872)	1
CAA	1	0.57%						
TGG			1	0.43%				

CS—patients with CA stenosis; CG—control group. ^a^
*p*-values were calculated with χ^2^ test. ^b^
*p*-values were calculated with Fisher’s exact test.

**Table 3 biomedicines-10-01912-t003:** Tth111I, N363S, and ER22/23EK genotypes in CS70 and CS90 groups.

Polymorphism	CS70 (*n* = 61)		CS90 (*n* = 56)		CS70 vs. CS90			
	*n*	%	*n*	%	*p* ^a^	Compared Genotypes or Alleles	OR (95% CI)	*p* ^b^
Tth111I genotype					0.351			
CC	32	52.46%	22	39.29%		CC + CT vs. TT	1.095 (0.194–6.198)	1
CT	25	40.98%	30	53.57%		CC vs. CT + TT	1.698 (0.768–3.8)	0.194
TT	4	6.56%	4	7.14%		CC vs. TT	1.446 (0.241–8.66)	1
Tth111I allele								
C	89	72.95%	74	66.07%		C vs. T	1.383 (0.762–2.52)	0.259
T	33	27.05%	38	33.93%				
N363S genotype								
AA	57	93.44%	53	94.64%		AA vs. AG + GG	0.808 (0.113–5.02)	1
AG	3	4.92%	3	5.36%				
GG	1	1.64%						
N363S allele								
A	117	95.90%	109	97.32%		A vs. G	0.645 (0.098–3.406)	1
G	5	4.10%	3	2.68%				
ER22/ER23EK genotype								
AG			1	1.79%				
GG	61	100.00%	55	98.21%				
ER22/ER23EK allele								
A			1	0.89%				
G	122	100.00%	111	99.11%				
Tth111I/N363S/ ER22/23EK haplotype					-			
CAG	84	68.85%	72	64.29%		CAG vs. other	1.227 (0.687–2.195)	0.490
TAG	33	27.05%	36	32.14%		TAG vs. other	0.784 (0.428–1.429)	0.473
CGG	5	4.10%	2	1.79%		CGG vs. other	2.342 (0.374–25.078)	0.449
TAA			1	0.89%				
TGG			1	0.89%				

CS70—patients with 70–89% stenosis; CS90—patients with 90% or more stenosis. ^a^ *p*-values were calculated with the χ^2^ test. ^b^ *p*-values were calculated with Fisher’s exact test.

**Table 4 biomedicines-10-01912-t004:** Tth111I, N363S, and ER22/23EK genotypes in CS&HT and CS&nHT.

Polymorphism	CS&nHT (*n* = 16)		CS&HT (*n* = 101)		CS&nHT vs. CS&HT			
	*n*	%	*n*	%	*p* ^a^	Compared Genotypes or Alleles	OR (95% CI)	*p* ^b^
Tth111I genotype					0.397			
CC	5	31.25%	49	48.51%		CC + CT vs. TT	1.116 (0.128–53.657)	1
CT	10	62.50%	45	44.55%		CC vs. CT + TT	0.485 (0.123–1.65)	0.281
TT	1	6.25%	7	6.93%		CC vs. TT	0.719 (0.064–38.552)	0.58
Tth111I allele								
C	20	62.50%	143	70.79%		C vs. T	0.689 (0.298–1.65)	0.408
T	12	37.50%	59	29.21%				
N363S genotype								
AA	15	93.75%	95	94.06%		AA vs. AG + GG	0.948 (0.103–46.446)	1
AG	1	6.25%	5	4.95%				
GG			1	0.99%				
N363S allele								
A	31	96.88%	195	96.53%		A vs. G	1.112 (0.135–51.751)	1
G	1	3.12%	7	3.47%				
ER22/ER23EK genotype								
AA								
AG			1	0.99%				
GG	16	100%	100	99.01%				
ER22/ER23EK allele								
A			1	0.50%				
G	32	100%	201	99.50%				
Tth111I/N363S/ ER22/23EK haplotype								
CAG	19	59.38%	137	67.82%		CAG vs. other	0.695 (0.304–1.631)	0.42
TAG	12	37.50%	57	28.22%		TAG vs. other	1.523 (0.635–3.524)	0.301
CGG	1	3.13%	6	2.97%		CGG vs. other	1.054 (0.022–9.15)	1
TAA			1	0.50%				
TGG			1	0.50%				

CS&HT—patients with CA stenosis and arterial hypertension; CS&nHT—normotensive patients with CA stenosis. ^a^ *p*-values were calculated with the χ^2^ test. ^b^ *p*-values were calculated with Fisher’s exact test.

**Table 5 biomedicines-10-01912-t005:** Tth111I, N363S, and ER22/23EK genotypes in CG&M and CS&M.

Polymorphism	CG&M (*n* = 35)		CS&M (*n* = 76)		CG&M vs. CS&M			
	*n*	%	*n*	%	*p* ^a^	Compared Genotypes or Alleles	OR (95% CI)	*p* ^b^
Tth111I genotype					0.913			
CC	16	45.71%	34	44.74%		CC + CT vs. TT	0.753 (0.137–5.144)	0.705
CT	16	45.71%	37	48.68%		CC vs. CT + TT	1.04 (0.429–2.505)	1
TT	3	8.57%	5	6.58%		CC vs. TT	0.788 (0.133–5.7)	1
Tth111I allele								
C	48	68.57%	105	69.08%		C vs. T	0.977 (0.51–1.9)	1
T	22	31.43%	47	30.92%				
N363S genotype								
AA	33	94.29%	72	94.74%		AA vs. AG + GG	0.917 (0.124–10.621)	1
AG	2	5.71%	3	3.95%				
GG	0	0.00%	1	1.32%				
N363S allele								
A	68	97.14%	147	96.71%		A vs. G	1.156 (0.184–12.428)	1
G	2	2.86%	5	3.29%				
ER22/ER23EK genotype								
AA	0	0.00%	0	0.00%				
AG	0	0.00%	0	0.00%				
GG	35	100.00%	76	100.00%				
ER22/ER23EK allele								
A	0	0.00%	0	0.00%				
G	70	100.00%	152	100.00%				
Tth111I/N363S/ER22/23EK haplotype								
CAG	46	65.71%	101	66.45%		CAG vs. other	0.968 (0.513–1.851)	1
TAG	22	31.43%	46	30.26%		TAG vs. other	1.056 (0.542–2.024)	0.876
CGG	2	2.86%	4	2.63%		CGG vs. other	1.088 (0.096–7.801)	1
TGG			1	0.66%				

CG&M—men without CA stenosis; CS&M—men with CA stenosis. ^a^ *p*-values were calculated with the χ^2^ test. ^b^ *p*-values were calculated with Fisher’s exact test.

**Table 6 biomedicines-10-01912-t006:** Tth111I, N363S, and ER22/23EK genotypes in CG&W and CS&W.

Polymorphism	CG&W (*n* = 53)		CS&W (*n* = 41)		CG&W vs. CS&W			
	*n*	%	*n*	%	*p* ^a^	Compared Genotypes or Alleles	OR (95% CI)	*p* ^b^
Tth111I genotype					0.944			
CC	24	45.28%	20	48.78%		CC + CT vs. TT	0.967 (0.134–6.096)	1
CT	25	47.17%	18	43.90%		CC vs. CT + TT	0.87 (0.355–2.129)	0.836
TT	4	7.55%	3	7.32%		CC vs. TT	0.902 (0.118–6.035)	1
Tth111I allele								
C	73	68.87%	58	70.73%		C vs. T	0.916 (0.463–1.795)	0.873
T	33	31.13%	24	29.27%				
N363S genotype								
AA	49	92.45%	38	92.68%		AA vs. AG	0.967 (0.134–6.096)	1
AG	4	7.55%	3	7.32%				
GG	0	0.00%	0	0.00%				
N363S allele								
A	102	96.23%	79	96.34%		A vs. G	0.969 (0.138–5.905)	1
G	4	3.77%	3	3.66%				
ER22/ER23EK genotype								
AA	0	0.00%	0	0.00%		AG vs. GG	1.561 (0.079–94.772)	1
AG	2	3.77%	1	2.44%				
GG	51	96.23%	40	97.56%				
ER22/ER23EK allele								
A	2	1.89%	1	1.22%		A vs. G	1.554 (0.08–92.968)	1
G	104	98.11%	81	98.78%				
Tth111I/N363S/ ER22/23EK haplotype								
CAG	68	64.15%	55	67.07%		CAG vs. other	0.879 (0.456–1.684)	0.758
TAG	32	30.19%	23	28.05%		TAG vs. other	1.109 (0.561–2.212)	0.872
CGG	4	3.77%	3	3.66%		CGG vs. other	1.033 (0.169–7.252)	1
TAA	1	0.94%	1	1.22%		TAA vs. other	0.773 (0.01–61.267)	1
CAA	1	0.94%						

CG&M—women without CA stenosis; CS&W—women with CA stenosis. ^a^ *p*-values were calculated with the χ^2^ test. ^b^ *p*-values were calculated with Fisher’s exact test.

**Table 7 biomedicines-10-01912-t007:** Associations of Tth111I, N363S, and ER22/23EK variants of NR3C1 gene and metabolic profile.

Parameter	Tth111I Genotype	N363S Genotype	ER22/ER23EK Genotype
BMI (kg/m^2^)	0.411	0.168	0.738
Dyslipidemia	0.475	0.325	0.584
DM	0.454	0.671	0.573
HT	0.496	0.741	1.0

DM—diabetes mellitus; HT—arterial hypertension. Data are given as mean *p*-values calculated with the Kruskal–Wallis test for quantitative variables and with the χ^2^ test for qualitative variables.

## Data Availability

The data presented in this study are available on request from the corresponding author.

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
