# Peer review of "Analysis of Genetic Variants in the Glucocorticoid Receptor Gene NR3C1 and Stenosis of the Carotid Artery in a Polish Population with Coronary Artery Disease"

_biomedicines, 2022, doi:10.3390/biomedicines10081912_

Round 1

Reviewer 1 Report

This paper is well written original and easy to read. The english in good. I think it can be of great scientific interest.

My minor point of revision:

“ Coronary artery disease (CAD), cerebrovascular disease, peripheral arterial disease, rheumatic heart disease, con-genital heart disease, deep vein thrombosis, and pulmonary embolism are classified as CVDs (3).”  Is redundant, I suggest to delete

“The mechanisms determining CVDs pathogenesis are extremely complex and include distinct interactions. Glucocorticoids (GCs) are steroid hormones synthesized and released by the adrenal cortex in a circadian manner and response to stress (10). They are associated with the development of CVDs (10). Thus, the therapeutic utilization of GCs is limited (11, 12)” need to be rephrased

This sentence is to long and very hard to read, please rephrase

Arterial hypertension (HT) was defined as either systolic blood pressure exceeding 140 mmHg, or diastolic blood pressure greater than 90 mmHg, or antihypertensive treatment; and diabetes mellitus (DM) as patients using antidiabetic medication or fasting plasma glucose higher than 125 mg% (6.9 mmol/l); dyslipidemia when at least one of the following criteria were met: low-density lipoprotein cholesterol (LDL-c) 115 mg% or higher, triacylglycerols (TG) 150mg% or higher, high-density lipoprotein cholesterol (HDL-c) below 40 mg% (men) or below 45 mg% (women). LDL-c, TG, and HDL-c were measured using enzymatic methods (Roche Diagnostics, Poland). BMI was calculated as weight/height2 [kg/m2].

Please rephrase

“The program enrolled 117 consecutive patients aged 56 - 90 (76 men and 41 women) with angiographically documented CAD admitted to the Clinic of Cardiology due to CA stenosis – CS group. The studied population was divided into group with 70% - 89% stenosis (CS70) including 61 patients (40 men and 21 women) and a group with a critical (≥90%) stenosis (CS90) including 56 patients (36 men and 20 women). All patients enrolled in the program underwent percutaneous carotid angioplasty with stent implantation (CAS).”

What about patients with CA plaque (stenosis < 70%)

Remove “one” in the :To the best of our knowledge, our study is the first one to

Author Response

Dear Sir/Madam,

Thank you for your engaging review. We revised our manuscript again and implemented all your adjustments to improve our paper. Below, we summarize modifications based on your review.

We deleted redundant sentence: “Coronary artery disease (CAD), cerebrovascular disease, peripheral arterial disease, rheumatic heart disease, congenital heart disease, deep vein thrombosis, and pulmonary embolism are classified as CVDs (3).”

We deleted redundant “one” from sentence: “To the best of our knowledge, our study is the first one to investigate the association between Tth111I, ER22/ER23 and N363S-NR3C1 gene polymorphisms and the CA stenosis.”

We rephrased:

  • previously: “The mechanisms determining CVDs pathogenesis are extremely complex and include distinct interactions. Glucocorticoids (GCs) are steroid hormones synthesized and released by the adrenal cortex in a circadian manner and response to stress (10). They are associated with the development of CVDs (10). Thus, the therapeutic utilization of GCs is limited (11, 12)”

currently: “Glucocorticoids (GCs) are one of the numerous factors determining CVDs patho-genesis. They are steroid hormones synthesized and released by the adrenal cortex in a circadian manner and as a response to stress. GCs are also used as therapeutic agents. Their side effects, like abdominal obesity, diabetes, and arterial hypertension (HT), are simultaneously CVDs risk factors (10-12).”

  • previously: “Arterial hypertension (HT) was defined as either systolic blood pressure exceeding 140 mmHg, or diastolic blood pressure greater than 90 mmHg, or antihypertensive treatment; and diabetes mellitus (DM) as patients using antidiabetic medication or fasting plasma glucose higher than 125 mg% (6.9 mmol/l); dyslipidemia when at least one of the following criteria were met: low-density lipoprotein cholesterol (LDL-c) 115 mg% or higher, triacylglycerols (TG) 150mg% or higher, high-density lipoprotein cholesterol (HDL-c) below 40 mg% (men) or below 45 mg% (women). LDL-c, TG, and HDL-c were measured using enzymatic methods (Roche Diagnostics, Poland). BMI was calculated as weight/height2 [kg/m2].”

currently: “HT was defined as either systolic blood pressure exceeding 140 mmHg, or diastolic blood pressure greater than 90 mmHg, or antihypertensive treatment. Subjects whose fasting plasma glucose was higher than 125 mg% (6.9 mmol/l) or those using antidiabetic medication were recognized as patients with diabetes mellitus (DM). Dyslipidemia was diagnosed when at least one of the following criteria was met:

  • low-density lipoprotein cholesterol (LDL-c) 115 mg% or higher,
  • triacylglycerols (TG) 150mg% or higher,
  • high-density lipoprotein cholesterol (HDL-c) below 40 mg% (men) or below 45 mg% (women).
  • LDL-c, TG, and HDL-c were measured using enzymatic methods (Roche Diagnostics, Poland). BMI was calculated as weight/height2 [kg/m2].”
  • previously: “The program enrolled 117 consecutive patients aged 56 - 90 (76 men and 41 women) with angiographically documented CAD admitted to the Clinic of Cardiology due to CA stenosis – CS group. The studied population was divided into group with 70% - 89% stenosis (CS70) including 61 patients (40 men and 21 women) and a group with a critical (≥90%) stenosis (CS90) including 56 patients (36 men and 20 women). All patients enrolled in the program underwent percutaneous carotid angioplasty with stent implantation (CAS).”

currently: “We defined case group (CS) as patients with angiographically documented CAD admitted to the Cardiology Clinic due to CA stenosis. The program enrolled 117 consecutive patients aged 56 - 90 (76 men and 41 women). This population was further divided into a subgroup with 70% - 89% stenosis (CS70), including 61 patients (40 men and 21 women), and a second subgroup with critical (≥90%) stenosis (CS90), including 56 patients (36 men and 20 women). All patients enrolled in the program underwent percutaneous carotid angioplasty with stent implantation (CAS).”

As mentioned in 2.1. Patients: in case group, all patients had either 70%-89% or ≥90% stenosis. In control group, ultrasound examination ruled out the presence of CA stenosis. We did not distinguish patients with CA stenosis <70% (CA plaque) because in our study there was no patient with CA plaque requiring percutaneous carotid angioplasty with stent implantation.

Best wishes!

Reviewer 2 Report

The manuscript is devoted to an actual topic, such as the significance of glucocorticoid receptor gene polymorphism in the development of carotid atherosclerosis. The discussion about the role of NR3C1 polymorphism in the development of cardiovascular pathology has been going on for many years, but the results of the study remain controversial and require clarification. The present study shows that Tth111I, N363S, and ER22/23EK-NR3C1 polymorphisms are not associated with carotid stenosis. This is a new result of a clarifying nature. The manuscript is well and clearly written, the discussion is detailed. Comment: The result obtained may be due to the fact that in the CS group, the number of men prevailed, whose susceptibility to vascular disorders and atherosclerosis is higher than that of women. It can be assumed that the influence of sex hormones can lead to a distortion of the results. It may be advisable to make calculations with a sex correction or to align the samples by sex.

Author Response

Dear Sir/Madam,

Thank you for your apt comment. We revised our manuscript following your guidance.

We conducted another analysis to exclude sex hormones’ influence. We compared men with stenosis to men without stenosis; moreover, we compared women with and without stenosis. None of these calculations led us to statistically significant results. They are summarized in the main text and included as two new tables: Tth111I, N363S, ER22/23EK genotypes in CG&M and CS&M; Tth111I, N363S, ER22/23EK genotypes in CG&W and CS&W.

Moreover, we improved some minor language lapses.

Best wishes!
